# Macrophage-Specific Coxsackievirus and Adenovirus Receptor Deletion Enhances Macrophage M1 Polarity in CVB3-Induced Myocarditis

**DOI:** 10.3390/ijms24065309

**Published:** 2023-03-10

**Authors:** Ha-Hyeon Shin, Eun-Seok Jeon, Byung-Kwan Lim

**Affiliations:** 1Department of Biomedical Science, Jungwon University, Goesan-gun 28024, Republic of Korea; 2Division of Cardiology, Samsung Medical Center, Sungkyunkwan University School of Medicine, 50 Irwon Dong, Gangnam-gu, Seoul 06351, Republic of Korea

**Keywords:** myocarditis, coxsackievirus and adenovirus receptor, coxsackievirus B3, macrophage polarity

## Abstract

The coxsackievirus and adenovirus receptor (CAR) is very well known as an epithelial tight junction and cardiac intercalated disc protein; it mediates attachment and infection via the coxsackievirus B3 (CVB3) and type 5 adenovirus. Macrophages play important roles in early immunity during viral infections. However, the role of CAR in macrophages is not well studied in relation to CVB3 infection. In this study, the function of CAR was observed in the Raw264.7 mouse macrophage cell line. CAR expression was stimulated by treatment with lipopolysaccharide (LPS) and tumor necrosis factor-α (TNF-α). In thioglycollate-induced peritonitis, the peritoneal macrophage was activated and CAR expression was increased. The macrophage-specific CAR conditional knockout mice (KO) were generated from lysozyme Cre mice. The expression of inflammatory cytokine (IL-1β and TNF-α) was attenuated in the KO mice’s peritoneal macrophage after LPS treatment. In addition, the virus was not replicated in CAR-deleted macrophages. The organ virus replication was not significantly different in both wild-type (WT) and KO mice at days three and seven post-infection (p.i). However, the inflammatory M1 polarity genes (IL-1β, IL-6, TNF-α and MCP-1) were significantly increased, with increased rates of myocarditis in the heart of KO mice compared to those of WT mice. In contrast, type1 interferon (IFN-α and β) was significantly decreased in the heart of KO mice. Serum chemokine CXCL-11 was increased in the KO mice at day three p.i. compared to the WT mice. The attenuation of IFN-α and β in macrophage CAR deletion induced higher levels of CXCL-11 and more increased CD4 and CD8 T cells in KO mice hearts compared to those of WT mice at day seven p.i. These results demonstrate that macrophage-specific CAR deletion increased the macrophage M1 polarity and myocarditis in CVB3 infection. In addition, chemokine CXCL-11 expression was increased, and stimulated CD4 and CD8 T cell activity. Macrophage CAR may be important for the regulation of innate-immunity-induced local inflammation in CVB3 infection.

## 1. Introduction

Coxsackievirus and adenovirus receptor (CAR) is a cell surface protein that mediates attachment and infection by coxsackievirus B3 (CVB3) and type 5 adenovirus. It is recognized as an epithelial tight junction and a cardiac intercalated disc protein [1,2,3,4]. CAR is highly expressed during embryonic and adult heart development, but is significantly reduced after birth and aging [5,6]. Also present in the brain and endothelial cells, the CAR of endothelial cells can be controlled by drugs [7,8]. CAR is a CVB3 receptor for induced myocarditis, suggesting that there is a close relationship between the heart and viral infection [9,10].

CVB3 is a non-enveloped virus containing a 7.4 kb positive-single-strand RNA genome that is an enterovirus in the picornaviridae family. It is the primary cause of viral myocarditis, which can induce serious inflammation in the immune system and develop into chronic myocarditis, dilated cardiomyopathy, and even heart failure and death [10,11,12]. CVB3 infection activates the innate immune response and induces the production of inflammatory cytokines, interleukin (IL)-1, IL-6, and tumor necrosis factor (TNF)-α in heart tissue [13,14,15,16]. During that time, the virus is harbored in the immune cells of lymphoid organs, temporarily escapes immune clearance, and is secondarily transported to other target sites such as the heart and pancreas [17]. Thus, immune cells such as those produced by macrophage viral infection may take advantage of CVB3 survival in the immune system [13,18,19]. Macrophage virus infection may abolish apoptosis and restricted cytopathic effect, which sustained the virus production. Knowledge in this area is interesting because it can lead to the basic understanding of many processes, such as virus-cell interactions that are receptor-based, cell-to-cell transmission upon infection, persistence, genetic changes of cell machinery, and therapy applications to certain cohorts of patients. Circulating monocytes and macrophages play one of the most important roles in the protection of the organism against viral infection. The causative virus entry and proliferation in the myocardium concludes with the activation of the host immune system, which attenuates viral proliferation but may also enhance viral entry [20,21].

Macrophages regulate the immune system against infectious pathogens such as bacteria and viruses, which secrete cytokines and chemokines with induced inflammation [22,23,24]. Macrophage polarization plays an important role in tissue repair and homostasis; it is characterized as the M1 type or the M2 type. The activation states and functions of macrophages can be classified as the M1 type (classically activated macrophages) and the M2 type (alternatively activated macrophages). Macrophage polarization is associated with responses to pro-inflammatory and anti-inflammatory reactions in various infectious diseases. The role of M1 macrophages is to secrete pro-inflammatory cytokines and chemokines such as IL-1β, IL-10, and TNF-α by TLR4 and T-helper cell-1 (Th1) secreting interferon-γ [25,26]. They participate in the positive immune response and function as an immune monitor [23]. At this time, when the M1-type activity is continued and matured, polarization occurs in the M2-type form, and the T helper cell 2 (Th2) is activated. M2 macrophages mainly secrete Arginase-I and IL-10, and other anti-inflammatory cytokines, which work to reduce inflammation and contribute to tumor growth, and have an immunosuppressive function. They are important in wound healing and tissue repair [22,26].

In this study, we examined the role of CAR in the macrophage by using the CVB3-induced myocarditis model. We found that macrophage-specific CAR deletion increased CVB3-induced myocarditis, nevertheless improved mice survival through prolonged M1 macrophage activity. These results demonstrate that macrophage-specific CAR deletion induced the M1 polarity macrophage and increased myocarditis in CVB3 infection. In addition, expression of the chemokine CXCL-11 is induced, and stimulates CD4 and CD8 T lymphocyte activity, which may have beneficial effects on mouse survival. Macrophage CAR deletion attenuated early macrophage activity, which delayed virus production and provided a recovery time to CVB3 infection mice.

## 2. Results

### 2.1. CAR Expression Was Increased in LPS and TNF-α Stimulated Macrophages

The expression of CAR in macrophages was observed in the Raw264.7 mouse macrophage cell line. The cell was stimulated by LPS (200 ng/mL) and tumor necrosis factor (TNF)-α (100 ng/mL) for 24 h. CAR expression was significantly increased by the LPS and TNF-α treatment (Figure 1A). For the LPS treatment, CAR expression was observed at two hours after LPS stimulation (Figure 1B). These observations focused on the macrophage activation related to CAR expression. To identify CAR induction from macrophages in the case of inflammatory conditions, CAR expression was confirmed using the thioglycolate-induced peritonitis model. Mice were intra-peritoneally injected with 1 mg/mL thioglycollate (Thio) for three days (*n* = 5 each group; control vs. thio). The mouse peritoneal macrophage (PMC) was then isolated and underwent the real-time PCR of CAR, interleukin (IL)-1b, IL-6, and TNF-a. The gene transcription of CAR, IL-1b, and TNF-a was significantly increased in the PMCs of peritonitis mice compared to wild-type mice (Figure 1C).

### 2.2. CAR Deficiency Increased Macrophage Inflammatory Cytokine Expression in CVB3 Infection

To identify the role of CAR in macrophages, we generated the conditional macrophage-specific CAR knockout mouse (KO) using CAR-floxed allele mice with lysozyme-Cre transgenic mice. The fluorescence-activated cell sorting (FACS) analysis of CAR and F4/80 expression confirmed CAR deletion using isolated PMC from wild-type (WT) and KO mice. We observed a lower percentage of CAR^+^ F4/80^+^ macrophages in KO compared to WT mice (2.59 ± 0.9% vs. 33.28 ± 1.2%; Figure 2A). We then compared the expression of the inflammatory cytokine IL-1b, IL-6, and tumor necrosis factor (TNF)-α in WT and KO mice PMCs using real-time PCR after stimulation with LPS (200 ng/mL). IL-1β, IL-6, and TNF-α were significantly induced in both WT and KO PMCs; however, levels of IL-1b and TNF-a were higher than that of KO PMC (Figure 2B). This may be due to CAR deletion attenuating LPS stimulating signaling induction. We then observed coxsackievirus B3 (CVB3) replication in both WT and KO PMCs. CVB3 replication was detected in WT PMC. However, when CAR was deleted from the PMC, no CVB3 viral protein production was detected for 120 h post-infection (p.i.) (Figure 2C). Moreover, CAR knockout macrophages showed higher induction levels of M1 polarity genes (IL-1b, MCP-1) than of M2 polarity genes (IL-10, Arg-1) in CVB3 infection compared to WT (Figure 2D).

### 2.3. Macrophage CAR Deficiency Improves Mice Survival in CVB3-Induced Myocarditis

We observed the effect of macrophage CAR deletion in CVB3-induced myocarditis (Figure 3A). The subacute phase mice survival rate was improved in KO mice compared to WT mice from three different repeat experiments (80 ± 9.2% vs. 51 ± 11%, *p* < 0.21; Figure 3B). In contrast, the heart and pancreas virus titer did not differ between WT and KO mice at day three p.i; however, heart virus replication was significantly increased in KO mice compared to WT at seven days p.i. (21,480 ± 8747 vs. 1643 ± 669 PFU/mg; Figure 3C). These results indicate a complicated relationship between improved mouse survival and increased virus replication in the heart from CVB3 infection.

### 2.4. Heart Inflammation Increased in CAR Knockout Mice

In terms of histology, KO mice showed a larger inflammation area compared to the WT mice. However, the damage to the myocardium did not differ between the groups according to the Evans Blue dye (EBD) uptake test (Figure 4A). The inflammatory lesion was significantly increased in KO hearts compared to WT hearts after day seven p.i (8 ± 1.2% vs. 2.8 ± 0.8%; Figure 4B). We hypothesized that such a decrease in CAR might be biologically important, and then tested this hypothesis. Using a real-time PCR, we observed the expression of pro- and anti-inflammatory cytokines in CVB3-induced-myocarditis. The macrophage M1 polarity gene (IL-1b, MCP-1) was significantly induced in the hearts of KO mice at day three p.i; however, the M2 polarity gene (IL-10 and Arg-1) was reduced in KO mice at days three and seven p.i. compared to WT mice (Figure 4C). These results suggest that macrophage CAR plays an important role in regulating the heart macrophage polarity and the induction of the inflammatory response.

### 2.5. CAR Deficiency Attenuated the Expression of Type1 Interferon

The macrophage CAR knockout showed far more significant protective effects on mouse survival in cases of CVB3-infection-induced inflammation and macrophage M1 polarity than were seen in WT mice. We investigated whether changes in the innate immune response and CAR ablation of macrophages affected the defense mechanisms used against viral infection. The type 1 interferon (IFN)-α, and β transcription levels did not differ between the WT and KO mice. However, the serum levels of IFN-α and β were significantly decreased in KO mice at day three p.i. (Figure 5A,B). Macrophage maturation was observed by F4/80, and CD68 mRNA real-time PCR. At day three of the CVB3 infection, the macrophage level did not differ between WT and KO mice. In contrast, macrophage localization was significantly increased in the hearts of KO mice compared to those of WT mice at day seven p.i; this result was similar to that for inflammatory cell infiltration in CVB3-infected mice hearts (Figure 5C). 

### 2.6. Macrophage CAR Ablation Induced T Cell Activation in Case of CVB3 Infection

Chemokine change was confirmed by the macrophage chemokine dot blot assay. Interestingly, the chemokine CXCL-11 and IL-16 were significantly changed in the serum three days p.i (Figure 6A,B). Macrophage CAR deficiency may attenuate CAR-induced macrophage stimulation in CVB3 infections, which may delay the activation of cellular immunity in CVB3-induced myocarditis. T-cell localization was confirmed by the immunohistochemistry of CD4 and CD8 T-cell marker expression. CD4- and CD8-positive T-cells were recruited at greater rates in the inflammatory areas of KO mice compared to WT mice hearts at day seven p.i (Figure 6C,D). These results indicate that T cells were more strongly activated in KO mice hearts compared to their WT counterparts. This might have a beneficial effect on mice survival in CVB3-induced myocarditis. Although we did not prove that macrophage-specific-CAR deletion has any direct effects, this experiment may support the novel finding that the virus reservoir macrophage can regulate inflammation and adaptive immune responses in CVB3 infection.

## 3. Discussion

To examine the role of macrophage CAR expression in the pathogenesis of myocarditis, we generated mice with CAR KO using the Cre-LoxP recombination system with the murine lysozyme promoter Cre. Peritoneal macrophages (PMC) were then isolated to determine whether CAR expression affects macrophage polarization by LPS treatment or CVB3 infection. Our initial observation was that macrophage M1 polarity was upregulated in the macrophage-specific CAR KO mice with CVB3-induced myocarditis. However, mice survival was improved, despite increased viral replication and inflammation lesions in the heart. This finding conflicts with previous results, raising the question of the mechanisms at work in these macrophage-specific CAR-deficient mice. The mechanisms involved in mediating the activity of the innate immune response remain unclear.

Our results indicate that CAR was stimulated by the LPS and TNF-α treated mouse macrophage 264.7 cell line. This result implies that CAR may work as a sensor of inflammation in the early immune response. M1 polarized macrophage gene expression was increased in the peritoneal macrophage (PMC) in the thioglycollate-induced peritonitis model. In contrast, CAR-deleted knockout isolated PMCs attenuated the expression of the proinflammatory cytokine genes (IL-1β, and TNF-α) with LPS treatment. This may be because they play important roles in regulating macrophage polarization in the early stage of LPS stimulation. In contrast, CVB3 infection led to M1 polarized gene expression even without CAR expression. To the best of our knowledge, this is the first macrophage-specific knockout approach that has been used to examine the role of macrophage CAR in CVB3 infection. Virus infection showed the opposite phenotypes to those seen in their activation states. CVB3 can infect PMC, but virus replication was delayed for 48 h in these cells, even though the macrophages did not show a cytopathic effect in CVB3 infection. CAR deficiency completely blocks CVB3′s entry into the PMC, but the macrophage is more stimulated, and the expression of the M1 polarity gene IL-1β and MCP-1 were significantly increased compared to WT. CAR deletion may affect the induction of innate immune defense mechanisms by allowing viruses sufficient time to activate macrophages. Macrophages attract other cells involved in the adaptive immune system, in particular T cells, to sites of chronic inflammation. Furthermore, macrophages can sense the time at which an injury is terminated and thus start the process of inflammation and control the healing phase [22]. Macrophages are the first subpopulation of immune cells that contact pathogens and can be infected and serve as a vehicle for virus dissemination; they are less likely to serve as a reservoir due to their naturally short life span and inability to support viral gene expression and replication [20,27].

Indeed, we previously found that CAR was up-regulated during inflammatory signaling induction in the heart, and identified macrophages as a novel cellular source of CAR in virus infection [9]. Further studies revealed that CAR is induced during monocyte-to-macrophage differentiation and is particularly strongly associated with M1 macrophages. By using animal models and viruses associated with lethal disease, we made the surprising discovery that these cells can be identified as part of a permissive system that assists the spread of viruses to target sites. Our results showed that when macrophages lose this function, viral replication and clearance from the target tissue was delayed. Notably, the subacute phase of heart inflammation was dramatically increased in KO mice, even though there were no differences in terms of the initial myocardium damage between WT and KO mice. Moreover, the serum levels of type 1 interferons were significantly decreased in KO mice at day three p.i, but those of mRNA were not; the matter of whether macrophage CAR expression regulates viral or non-viral pathogen immune responses is a fruitful topic for further study.

Finally, we confirmed the serum chemokine expression pattern by using a chemokine multi-dot blotting system. CXCL-11 expression was significantly increased in KO mice infected with CVB3; this chemokine is a well-known T-cell activator. The induction of CXCL-11 likely triggers the adaptive cellular immunity of CD4+ and CD8+ T cells [28]. Our results showed that CD4+ and CD8+ T cells were recruited at greater rates in the inflammatory lesions of the heart at day seven p.i. This might be the main reason for improved subacute phage survival in CVB3-induced myocarditis. However, other mechanisms may also be important for CAR regulation during macrophage polarization. More studies are needed to elucidate the complicated relationship between delayed virus replication with increased inflammation and improved survival in CVB3 infection. We thought that the CAR deletion-induced macrophage polarities might have a different pattern of immunity regulation compared with virus attachment to the CAR on the surface of wild-type macrophages. Without virus infection, prolonged macrophage activity could protect the inflammatory cytokine-induced cytotoxicity and direct virus toxicities. Our data showed that the mortality of the subacute phage (after five days of virus infection) dramatically decreased in CAR-KO mice. Delayed early-time virus replication and inflammatory immune response are unusual patterns in CVB3 disease, which could benefit virus clearance from the body and support mice recovery through T lymphocyte activity after seven days post-infection. The mice will survive when they overcome the subacute phase virus infection (Figure 7). This study showed that the coxsackievirus receptor CAR is upregulated in macrophages under inflammatory cytokine treatment. CAR deficiency in macrophages is specifically associated with M1 polarization and myocarditis in CVB3 infection. These results provide new insights into how CAR in macrophages regulate the innate and adaptive immune response in viral infections.

## 4. Materials and Methods

### 4.1. Viruses and Cells

Coxsackievirus B3 (CVB3) was derived from the infectious cDNA copy of the cardiotrophic CVB3-H3, and it was amplified in HeLa cells [15]. The virus-infected HeLa cells and tissue virus titer were determined with a plaque-forming unit (PFU) assay [29]. Raw264.7 cells and peritoneal macrophage cells (PMC) were cultured in Dulbecco’s Modified Eagle’s Medium (DMEM, Welgene, Inc., Gyeongsan-si, Korea) supplemented with 10% fetal bovine serum (FBS) and 1% penicillin-streptomycin solution (Welgene, Inc.) at 37 °C.

### 4.2. Isolation of Peritoneal Macrophage Cells (PMCs)

Peritoneal macrophages from WT mice and KO mice were collected (in sterile phosphate-buffered saline (PBS)) from the peritoneal cavities of unstimulated (*n* = 3) or thioglycollate-stimulated (*n* = 3) mice. The cells were washed, centrifuged at 4 °C 2000 rpm for 10 min, re-suspended 3-times in PBS, and plated on six-well and twelve-well culture plates in DMED supplemented with 10% fetal bovine serum and 1% PS complex. After being incubated for 6 h at 37 °C in 5% CO_2_, the adherent peritoneal macrophage was washed twice with DMEM to remove all of the non-adherent cells. Following incubation in an incubator for 72 h at 37 °C in 5% CO_2_, the cells were treated with stimulation. All experiments were repeated three times, and each separate experiment’s data was presented as a mean value.

### 4.3. Generation of Macrophage-Specific CAR Knockout Mice and CVB3 Infection

Conditional CAR knockout mice with the “floxed” CAR allele were generated as reported previously [1]. To generate macrophage-specific-conditional CAR knockout mice, CAR “floxed” allele mice were crossed with murine M lysozyme-Cre transgenic mice [30]. At 3 weeks after birth, genotype PCRs were derived using the genomic DNA from the pups’ tail extracts. Mice were housed at the Jungwon University animal facility. Dimmers were used in the mice’s rooms to create twilight periods between the light and dark cycles. A room temperature between 20–and 26 °C was maintained. The myocarditis model study was performed as described previously. The six-week-old MacKO (f/f Cre, KO, *n* = 30) mice and their littermate wild-type (f/f or Cre only, WT, *n* = 30) male or female mice (about 20 g in weight) were infected intraperitoneally with 2 × 10^5^ plaque-forming units (PFU) of CVB3. Mice were sacrificed on day 3 (*n* = 15) and day 7 (*n* = 15) post-infection from three separate experiments, and then the hearts, livers, and pancreas were harvested and analyzed. Prior to analysis, mice were injected with Evans blue dye 14 h before being sacrificed. Heart inflammation and myocardium damage were observed by histologic analysis. All procedures were reviewed and approved by the Institutional Animal Care and Use Committee of the Samsung Biomedical Research Institute (SBRI, #20191227002). SBRI is accredited by the Association for Assessment and Accreditation of Laboratory Animal Care International (AAALAC International) and abides by the Institute of Laboratory Animal Resources (ILAR) guidelines.

### 4.4. Total RNA Extraction and Quantitative Real-Time PCR

RNA was isolated from virus-infected mice hearts. Total RNA was extracted using TRIzol^®^ reagent (ThermoFisher Scientific, Cambridge, MA, USA) according to the manufacturer’s instructions. For RNA quantification, we synthesized complementary DNA (cDNA) using 1 µg RNA through a reverse transcription reaction using an oligo-dT primer. Real-time PCR quantitative RNA and DNA analyses were performed with an ABI Sequence Detection System using the SYBR green fluorescence quantification system (Applied Biosystems, Waltham, MA, USA). The standard PCR conditions were 95 °C for 10 min, then 40 cycles at 95 °C (30 s), and 60 °C (60 s), followed by a standard denaturation curve. All samples were tested in duplicate. The primer sequences are provided in Table 1.

### 4.5. Western Blot Analysis

Protein was extracted from WT and KO mice heart at 3 and 7 days after infection with CVB3, and peritoneal macrophage cells were separated on 10% SDS-PAGE, transferred to nitrocellulose membranes, and immunoblotted in 5% skim milk using standard methods. The membranes were probed with the primary antibodies anti-CAR, CVB3 VP1 (ThermoFisher Scientific), F4/80, CD68, and GAPDH (Cell Signaling, Danvers, MA, USA). Detection was performed using an ECL solution (Intron Biotech, Inc., Seongnam-si, Korea), and band intensities were quantified using ImageJ 1.53e software [31].

### 4.6. Histopathology and Immunohistochemistry

The hearts, pancreas, and liver were fixed in 10% formalin and embedded in paraffin; hematoxylin and eosin (H&E) staining was performed using 10-μm paraffin-embedded sections as described previously [31]. Inflammatory cell infiltration was observed under a light microscope. CD4- and CD8-positive T lymphocyte localization were observed by immunohistochemistry (IHC) staining. Paraffin-embedded sections of WT and KO mice hearts were stained with anti-mouse CD4 and CD8 antibodies (ThermoFisher Scientific) for 18 h at 4 °C. Immunodetection was performed using a Universal Quick Kit (Vector Laboratories, Burlingame, CA, USA) as described in the manufacturer’s instructions. Images of the stained tissue were taken and processed using a light microscope (Olympus Co., San Jose, CA, USA). 

### 4.7. Statistical Analysis

All data were analyzed using Prism 3 software (GraphPad Software, San Diego, CA, USA) and are presented as the means ± standard error of the mean (SEM). For analysis of the statistical significance between the two groups, a Student’s *t*-test was used, and a one-way ANOVA was used to analyze the statistical significance between multiple groups. For the analysis of multiple-time-point experiments, a two-way ANOVA was used. The survival rates of mice were analyzed by the Kaplan–Meier method. * *p* <0.05, ** *p* <0.01, *** *p* < 0.001 was considered significant.

## Figures and Tables

**Figure 1 ijms-24-05309-f001:**
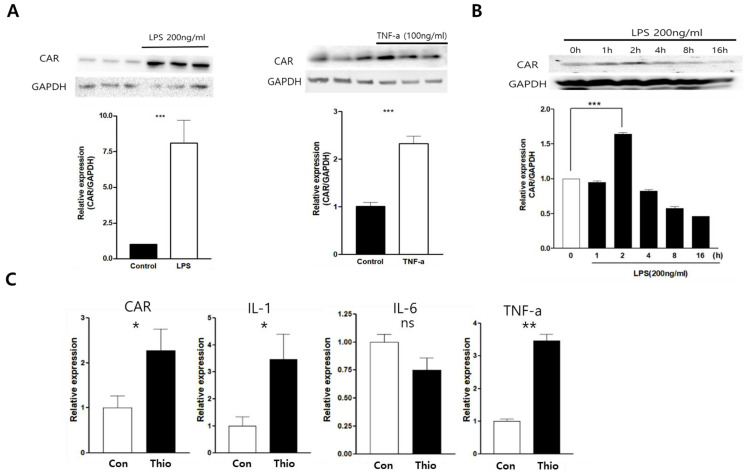
CAR expression was increased in LPS and TNF-α-stimulated macrophages. The Raw264.7 murine macrophage cell line was stimulated by LPS (200 ng/mL) and TNF-α (100 ng/mL) for 24 h, and protein and RNA were extracted. (**A**) Protein was subjected to western blot analysis using the indicated antibodies. (**B**) CAR bands indicate fold changes in CAR, which was normalized to GAPDH bands. A time-dependent change in CAR expression was shown, and the fold changes in CAR were normalized to GAPDH. (**C**) A thioglycollate-induced peritonitis mice model was generated, and the peritoneal macrophage cell (PMC) was isolated. The animal experiment was repeated three times. CAR, IL-1b, IL-6, and TNF-a mRNA levels were confirmed by real-time PCR. All data are expressed as the mean ± SEM from three independent experiments. NS, not significant, * *p* < 0.05, ** *p* < 0.01 and *** *p* < 0.001 according to a two-tailed Student’s *t*-test.

**Figure 2 ijms-24-05309-f002:**
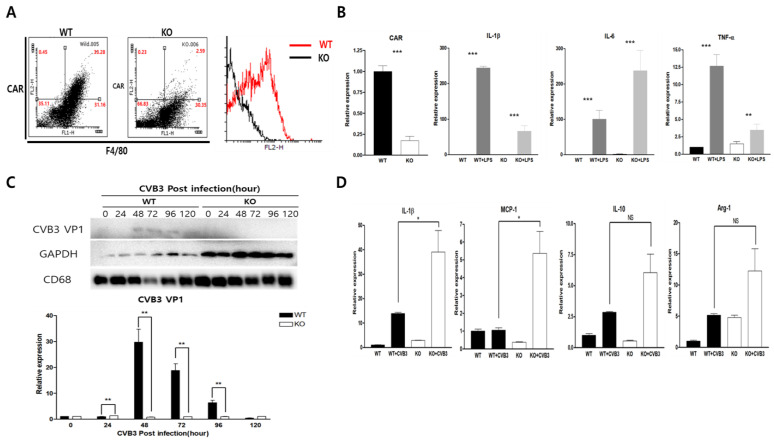
Generation of macrophage-specific CAR knockout mice. (**A**) PMC was isolated from wild-type (WT) and CAR knockout (KO) mice. The PMC was subjected to immunofluorescent staining using indicated antibodies and then subjected to FACS analysis. (**B**) RNA was extracted from WT and KO mice PMCs after LPS stimulation, after which CAR, IL-1beta, IL-6, and TNF-alpha mRNA levels were confirmed by real-time PCR. (**C**) WT and KO mice were infected with CVB3. The protein was extracted in a time-dependent manner and then subjected to western blot analysis using the indicated antibodies. (**D**) The mRNA levels of M1 (IL-1b and MCP-1) and M2 (IL-10 and Arg-1) polarity genes were confirmed by real-time PCR from CVB3-infected WT and KO PMCs. All data are expressed as the mean ± SEM from independent repeat experiments. NS, not significant, * *p* < 0.05, ** *p* < 0.01 and *** *p* < 0.001 according to the two-tailed Student’s *t*-test.

**Figure 3 ijms-24-05309-f003:**
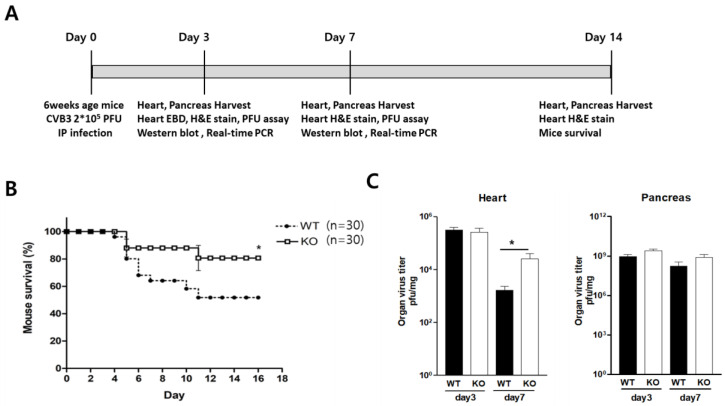
Macrophage-specific CAR deletion increased tissue virus replication but improved mouse survival. (**A**) CVB3 was injected intraperitoneally into WT and KO mice. (**B**) The mouse survival rate was recorded at day 14 p.i. (WT *n* = 30, KO *n* = 30). (**C**) The heart and pancreas were lysed on day three and day seven p.i. Tissue lysates were subjected to a plaque-forming unit (PFU) assay to measure the tissue virus titers. All data are expressed as the mean ± SEM. * *p* < 0.05, according to the two-tailed Student’s *t*-test.

**Figure 4 ijms-24-05309-f004:**
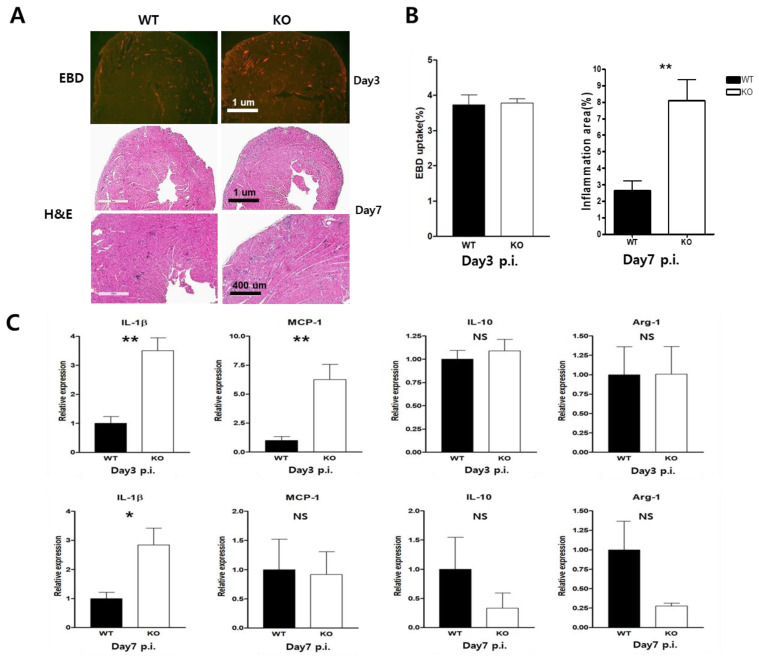
Heart inflammatory cell infiltration increased in CAR knockout mice. (**A**) WT and KO mice were injected with 10% Evans blue dye (EBD) for 18 h before being sacrificed. The heart was sectioned and subjected to fluorescence microscopy to determine the area of EBD uptake at day three p.i. The histological findings from the sectioned hearts showed inflammatory cell infiltration using Hematoxylin and Eosin (H&E) staining at day seven p.i. (**B**) Quantification of EBD uptake lesions and inflammation lesions (%) of the heart (*n* = 3 for each group). (**C**) RNA extracted from the heart was subjected to real-time PCR using the indicated primers. All data are expressed as the mean ± SEM. NS, not significant, * *p* < 0.05 and ** *p* < 0.01 according to a two-tailed Student’s *t*-test.

**Figure 5 ijms-24-05309-f005:**
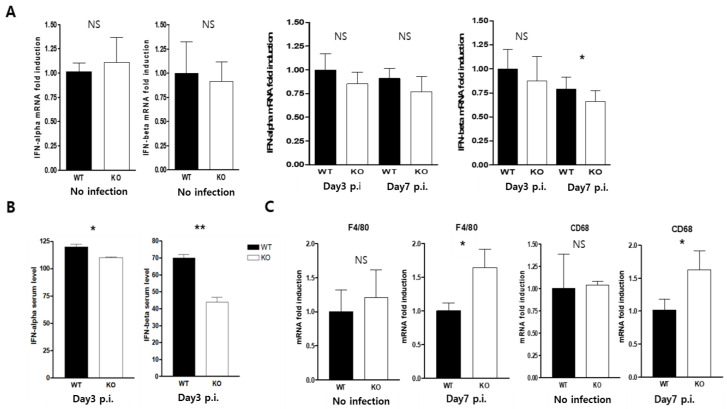
CAR deletion suppressed the serum levels of type 1 interferon levels in CVB3 infection. (**A**,**C**) Total RNA was extracted from WT and KO mice hearts at day three and seven p.i, and then subjected to real-time PCR with the indicated primer sets. (**B**) The serum levels of IFN-α and IFN-β were measured using the ELISA system. All data are expressed as the mean ± SEM. NS, not significant, * *p* < 0.05 and ** *p* < 0.01 according to a two-tailed Student’s *t*-test.

**Figure 6 ijms-24-05309-f006:**
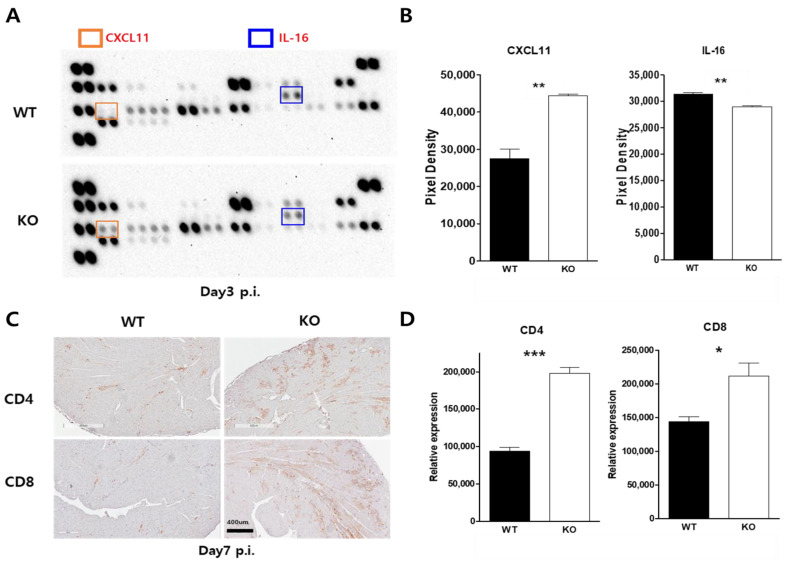
Macrophage CAR deficiency induced T-cell activation in CVB3 infection. (**A**) The serum was collected from WT and KO mice three days p.i. and then subjected to a chemokine multi-dot analysis system. (**B**) The chemokine dot density was quantified using ImageJ software. (**C**) Hearts were sectioned and subjected to immunohistochemistry. Brown stains indicate CD4- and CD8-positive T lymphocytes in WT and KO mice hearts at day seven p.i. (**D**) Positive cells were quantified by ImageJ software. All data are expressed as the mean ± SEM from independent experiments. * *p* < 0.05, ** *p* < 0.01 and *** *p* < 0.001 according to a two-tailed Student’s *t*-test.

**Figure 7 ijms-24-05309-f007:**
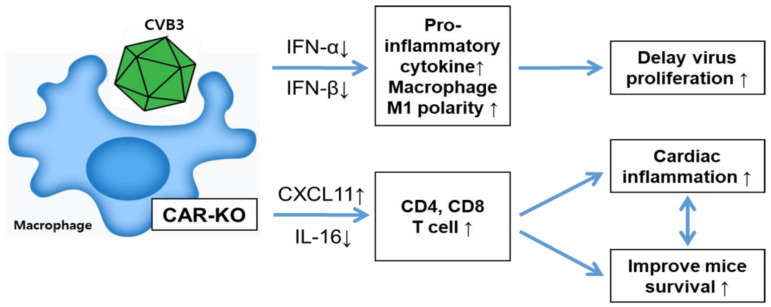
A proposed model of the mechanism of macrophage-specific CAR deletion in CVB3 infection. CVB3 binds with its receptor and initiates an early innate immune response. Viral infection induces macrophage M1 polarity and CXCL-11 expression, which upregulates the induction of CD4- and CD8-positive T lymphocytes into cardiac focal inflammatory lesions. This results in positive feedback that increases M1 macrophage activation and delays viral replication. T-cell activation might have the beneficial effect of maintaining the subacute phase of mice survival.

**Table 1 ijms-24-05309-t001:** Real-time PCR primer sequences.

	Sense (5′→3′)	Antisense (5′→3′)
CAR	GGA CTA CTT GCA CTC CGA GAA G	CAT AGT GGC ACC GTC CTT GAT C
MCP-1	ACC TGG ATC GGA ACC AAA TG	CCT TAG GGC AGA TGC AGT TTT AA
IL-1β	TTG ACG GAC CCC AAA GAG TG	ACT CCT GTA CTC GTG GAA GA
IL-6	GTA CTC CAG AAG ACC AGA GG	TGC TGG TGA CAA CCA CGG CC
TNF-α	TTG ACC TCA GCG CTG AGT TG	CCT GTA GCC CAC GTC GTA GC
IFN-α	GCA ATG ACCATCC ATC AGC AGC T	GTG GAA GTA TGT CCT CAC AGC C
IFN-β	GCC TTT GCC ATC CAA GAG ATG C	ACA CTG TCT GCT GGT GGA GTT C
IL-10	AGT GAA CTG CGC TGT CAA TG	TTC AGG GTC AAG GCA AAC TT
Arg-1	CGC CTT TCT CAA AAG GAC AG	CGC CTT TCT CAA AAG GAC AG
CD4	GTT CAG GAC AGC GAC TTC TGG A	GAA GGA GAA CTC CGC TGA CTC T
CD8	ACT ACC AAG CCA GTG CTG CGA A	ATC ACA GGC GAA GTC CAA TCC G
F4/80	GGA AAG CAC CAT GTT AGC TG	CCT CTG GCT GCC AAG TTA AT
CD68	CTT CCC ACA GGC AGC ACA G	AAT GAT GAG AGG CAG CAA GAG G
GAPDH	ATC AAC GAC CCC TTC ATT GAC C	CCA GTA GAC TCC ACG ACA TAC TCA GC

## Data Availability

All relevant summary data are within this article.

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
