# Peer review of "Macrophage-Specific Coxsackievirus and Adenovirus Receptor Deletion Enhances Macrophage M1 Polarity in CVB3-Induced Myocarditis"

_ijms, 2023, doi:10.3390/ijms24065309_

Round 1

Reviewer 1 Report

Despite the interesting topic discussed in the proposed article and usage of advanced experimental techniques, there are some drawbacks.

In the materials and methods section, it would also be desirable to indicate the total number of animals used and provided more clear explanation of three experiments done.

The number of samples (n=3) shown in Figure 1, 4, 5  seems too small to objectively assess the effect, and the results in parts d,e,f do not convince of an adequate and correct analytical approach. Although the number of n=3 includes three separate experiments (it would be desirable to indicate the number of animals used in each separate experiment as well), the arithmetic mean value (obtained from the average of three experiments) is not convincing as a correct indicator of central tendency. Especially if the data dispersion (SD) is not indicated, but the SEM is used, which is large enough to create complete confidence in the impossibility of the significance of the differences at such a minimal number of n (contrary to what the authors presented and claimed). Since the presented mean value with  dispersion SD (calculated from provided SEM) contradict the basic postulates of biostatistics, which states that in a normal distribution, 68% of data should be within 1SD, 95% of data should be within 2SD, and 99.7% should be within 3SD, these results should be revised. Especially, because by modelling of presented relative expression (randomly generating mean data with SD from Figure 1c - CAR, IL-1) has been received negative values, which is a complete alogism and nonsense. The same is concerning Figure 4c (MCP-1, IL-10, Arg-1) and Figure 5c (F4/80, CD68). 

Usage of SEM is old-fashion approach, much more informative is scatter graph with represented SD. In case of an uneven or asymmetric distribution, the median with IQR should be applied

The quality of the bar graphs is quite poor: the font size is too small, as well different, individual graphs are compressed or stretched, which makes it difficult to perceive the displayed information. Therefore, regarding fonts and size Figures must be equalised.

A thorough revision of the manuscript and correction of all these deficiencies is recommended.

The scientific value of the proposed manuscript would be better if the fundamental error related to the non-observance of one of the basic rules of statistical analysis on the distribution of data and the choice of the correct method for determining the average feature, otherwise the also low-quality presented results are misleading to the reader and reduce the value of this research work.

Author Response

Thank you for your overall positive review of our manuscript, "Macrophage-specific Coxsackievirus and Adenovirus Receptor deletion enhances macrophage M1 polarity in CVB3-induced myocarditis" We are pleased with your comments that have improved our manuscript. Please find a point-by-point response to the review. We appreciate your reconsideration of this manuscript.

Reviewer 2 Report

This study investigated the role of the macrophage CAR in coxsackievirus B3 (CVB3)-induced myocarditis. The authors demonstrated the expression of CAR in mouse macrophages, which could be further induced in vitro and in vivo under inflammatory stimulation. The authors observed that gene expression of IL-1b and TNF-a was reduced in macrophages of CAR KO mice following LPS treatment compared to WT-mice. Furthermore, the authors reported an increase in the expression of M1 polarity genes and more severe myocarditis and a reduction in type-I IFN gene expression in the heart of CAR-KO mice compared to WT mice, which were accompanied by increased CXC-11 level and T cell activation. It was concluded that macrophage-specific CAR deletion increases the macrophage M1 polarity and myocarditis, and that macrophage CAR plays a key role in controlling innate immune response to the virus in the heart.  

Although the observations made in this study are interesting, the current research is mostly descriptive, lacking mechanistic insights. This is a major concern. For example, mechanism by which IL-1b and TNF-a are downregulated in macrophage of CAR KO mice treated with LPS has not been addressed/discussed. In addition, what is the mechanism that CAR KO macrophages exhibit more M1 than M2 phenotype? The results in CVB3-infected mouse model are very confusing, lacking in depth investigation of the underlying mechanisms for the discrepancy between mouse survival, tissue injuries/inflammations, and viral replication.  

Fig. 1, whether CVB3 infection induces CAR expression?

The summary figure of Figure 7 is not fully supported by the data.

There are numerous typos and errors in English grammar and syntax. Please pay attention to them.

Author Response

(The authors gave the same response as above.)

Reviewer 3 Report

Shin et al presented the manuscript and studied the role of CAR in murine macrophages during CVB3 infection. The data presented showed correlation but not causation and did not agree with the conclusion in the manuscript. The current manuscript is not well written and should not be accepted in this journal. Please find my specific points below.

1. The authors generated Lys-Cre CAR mice and infected them with CVB3. Does deletion of CAR in macrophages impair macrophage viability or cell numbers or function? Why the trend of IL-6 and TNF-a is different in KO cells when stimulated with LPS?

2. The data presented here are either not correctly interpreted. For example, p value in Figure 3B <0.21 doesn't mean significance. It is also inappropriate to use student t-test to evaluate the group difference in a survival curve. The authors showed increased CD4/CD8 T cell staining by IHC in Figure 6C, which only suggested more CD4/CD8 T cells rather than T cells were more activated. More CXCL11 and T cells are correlation, which doesn't necessarily suggest CXCL11 induced more T cells. I did not agree with the authors on the improvement of survival after CAR is deleted in mouse macrophages. 

3. The overall data presentation is not good. House keeping proteins in many WB blots are not even. Bar graphs should be presented with individual dots. Macrophage maturation should be measured by CD68 proteins instead of mRNA. The difference in type I IFN is marginal. All the Greek letters are not properly shown in the text. English needs great improvement as multiple typos are seen. 

Author Response

Thank you for your valuable comments and suggestion. Our study has a lot of weak points. However, your comments will improve our manuscript, and we could raise new insight related to macrophage polarity control and cellular immunity cytotoxicity. Please find a point-by-point response to the review. We appreciate your reconsideration of this manuscript.

Round 2

Reviewer 1 Report

I do accept improvements done by authors, but despite that, I still rate the work generally as moderately satisfactory.

Author Response

  • I appreciate your consideration. I confirmed your previous recommendation again and try to make a better modification to satisfy. However, there is a limitation to fixing all following your order. Please I would like to ask your favor about reconsidering our manuscript.

Reviewer 2 Report

The authors should at least discuss about the potential mechanisms underlying their observations in the revision.

Author Response

  • I agree with your comment. I responded, "The CVB3 infection model showed an obvious phenotype. Virus infection induces tissue injury and inflammation with virus replication. In addition, mouse mortality is about 50%. However, CAR KO mice were observed to improve survival. I thought that this might be an interesting point. More inflammation is due to M1 polarized macrophage, which benefits mice survival in acute phase viral infection. Prolonged macrophage activity could protect the second immune response-induced cytotoxicity." In the discussion, I have mentioned the potential mechanism of macrophage CAR-KO mice survival improvement.

"We thought that the CAR deletion-induced macrophage polarities might have a different pattern of immunity regulation compared with virus attachment to the CAR on the surface of wild-type macrophage. Without virus infection, prolonged macrophage activity could protect the inflammatory cytokine-induced cytotoxicity and direct virus toxicities. Our data showed that the mortality of subacute phage (after five days of infection) dramatically decreased in CAR-KO mice. Delayed early-time virus replication and inflammatory immune response are unusual patterns in CVB3 disease, which could benefit virus clearance from the body and support mice recovery through T lymphocyte activity after seven days post-infection. Because the mice will survive when they overcome the subacute phase virus infection.”

Reviewer 3 Report

Really appreciate the author’s efforts to improve the manuscript. However, the point-by-point responses are not satisfactory. For example, the authors added more mice to the cohort and the statistical analysis’ showed p<0.05. The authors implement more data to a 16-day experiment within 10 days receiving my report. Seems fishy to me. T test shouldn’t be used for this type survival study. 

Author Response

  • First of all, I have to apologize for my unfindable data explanation. I understand that I did not give any reason for the data change. It is not the additional experiment data. I have added the sacrificed mice number, which was not included in the previous survival data of total mice numbers. The reason why mice mortality was a little increased in WT mice. When sampling the time course, we should sacrifice more sick mice first. These mice are not counted as dead. The survival rates of mice were analyzed by the Kaplan–Meier method. I added it to the methods section.

Round 3

Reviewer 3 Report

The authors have addressed the concerns. 

Author Response

Thank you for your valuable comments and criticism. It is helpful to improve this manuscript and my research. I have rechecked the typo error.